# Comparative Study on Numerical Calculation of Modal Characteristics of Pump-Turbine Shaft System

**Xuyang Liu** [1], **Jiayang Pang** [2], **Lei Li** [1], **Weiqiang Zhao** [3], **Yupeng Wang** [1], **Dandan Yan** [2], **Lingjiu Zhou** [2] and **Zhengwei Wang** [3,*]

1   Henan Luoning Pumped Storage Co., Ltd., Luoning 471700, China
2   College of Water Resources and Civil Engineering, China Agricultural University, Beijing 100083, China; pangjy9603@163.com (J.P.); zlj@cau.edu.cn (L.Z.)
3   Department of Energy and Power Engineering, Tsinghua University, Beijing 100084, China; zhaoweiqiang@mail.tsinghua.edu.cn
*   Correspondence: wzw@mail.tsinghua.edu.cn; Tel.: +86-136-0136-3209

**Abstract:** Because a pump-turbine mainly undertakes the role of energy conversion and pumped storage in the field of hydropower engineering, the complex transition process and frequent conversion between different working conditions lead to the increase in the stress and strain of core components such as the unit shaft system, and even cause resonance phenomena. Based on ANSYS finite element numerical calculation software, this paper adopts the acoustic fluid–structure coupling method to study the influence of the shaft of the pump-turbine on the dynamic characteristics of the runner. At the same time, the paper analyses the influence of different contact modes between the runner and the shaft on the modal characteristics of the shaft system. The numerical simulation results have shown that the runner is affected by the added mass of the water. The natural frequency reduction rate of each order of wet modal is ranged from 19% to 64%. The main shaft has a greater influence on the simplification of the shaft system calculation method. The type of contact surface between the main shaft and the runner has a smaller influence on the modal characteristics and the natural frequency of the shaft system. The research in this paper contributes an evaluation of the dynamic characteristics of the runner of a hydraulic turbine unit, which is of great significance for the optimization of the analysis algorithm of the runner structure for large pumped storage units.

**Keywords:** pumped storage power station; numerical calculation; modal analysis; natural frequency

## 1. Introduction

Marine renewable energy is a kind of clean, environmental protection, with abundant reserves of energy, no pollution, high reserves of significant characteristics, and has become one of the most promising energy prospects. Since the 1970s, marine renewable energy has attracted wide attention in countries around the world. In the 21st century, the global energy demand has increased significantly, the shortage of oil, coal and other fossil fuels has intensified, and energy conservation and emission reduction are under great pressure to address global climate change. Therefore, the international community has reached a consensus on the strategic position of marine renewable energy in the future energy field. A marine pumped storage power station is the most reliable, most economic, long life cycle, large capacity, and most mature technology energy storage device in the power system, and is an important part of the development of new energy [1]. Pumped storage technology, as an important means of energy storage and transformation, can store unstable tidal energy and offshore wind energy, which has a strong peaking effect on the power grid, and is of great significance for the stable operation of the power grid and ensures a stable supply of energy [2].

The pumped storage power station has many uses, such as peak load and valley filling, frequency and phase regulation and emergency backup, and plays an increasingly

important role in optimizing the energy structure, promoting the development and utilization of new energy and protecting the ecological environment. The natural frequency and vibration mode of the runner in water are important parameters in the design of the runner [3]. By accurately estimating the frequency and vibration mode of the runner, effective measures are taken to avoid resonance or beat vibration of the runner and avoid rapid fatigue damage of the runner. In the actual running process, the runner is surrounded by water, and the vibration of the runner structure will drive the surrounding fluid to vibrate together, so that the fluid will produce some additional mass, which changes the modal characteristics of the runner [4].

Therefore, many experts and scholars around the world have paid attention to and studied the performance of the runner structure of the hydro-generator set. Egusquiza et al. [5] performed a harmonic response analysis on the runner of the pump-turbine in order to determine the cause of the runner's failure, and applied harmonic excitation simulating pressure pulsation to the runner, and found that the structural response depended on the excitation force vibration type and the structural natural frequency vibration type. Seidel et al. [6] measured the strain of a Francis turbine runner by arranging strain gauges. Based on the strain gauge database, the calibration method of RSI stress is established and optimized. This method can optimize the dynamic stress and fatigue of the mixed flow runner in the range of middle and high head. Based on the unidirectional fluid–structure coupling method, Negru et al. [7] studied the runner stress caused by steady flow and determined the dangerous position, and plotted the distribution diagram of the pressure coefficient on the blade to evaluate the blade load and the area with cavitation risk. Guillaume et al. [8] used the fluid–structure coupling method to analyze the influence of static and static interference on the dynamic stress and its performance of the runner, and evaluated the correlation between the directional components of the pulsating pressure and the dynamic stress of the runner. Rodriguez et al. [9] used a modal analysis method to conduct experimental research on the turbine runner model. The runner was freely suspended in air and water for multiple impact tests, and the model's natural frequency, damping ratio and vibration mode were obtained. The runner's vibration mode in water and air was the same, but the natural frequency was lower and the damping ratio was higher. This difference depends on the additional mass effect and modal shape of the water, not on the additional damping of the water [10]. Lais et al. [11] used two methods to study the modal characteristics of the real and model runner, and analyzed the effects of different materials, different model sizes and different hub geometries on the natural frequency, vibration pattern and frequency drop rate in water of the runner. Presas et al. [12,13] further improved the test method by changing the traditional hammer method to piezoelectric plate excitation, and verified the feasibility of the method. The model runner of the pump-turbine was tested by this method, and the influence of the shaft and volute on the modal characteristics was analyzed. Xu [14] introduced a nonlinear modal method based on the finite element method to analyze the dynamic interaction between the subsystems of a Francis turbine, combined with the influence of fluid–structure coupling. Based on the coupling effect, Shi [15] proposed a mathematical model of bent–torsional coupling vibration of an unbalanced rotor for turbine units. Hoerner [16] studied the fluid–structure coupling of the flexible blades of axial-flow turbines by combining numerical simulation and experimental research. Zhang [17] conducted fluid–structure coupling on real three-dimensional blades of a Francis turbine, and the numerical results show that the flow distribution in the flow channel is greatly affected by blade curvature and blade vibration, among which vibration has a significant effect on the near-wall flow structure. Based on the theory of fluid–structure coupling, the static stress distribution of a mixed-flow runner is calculated, the correlation between the stress and flow rate and the head is explored, and the possibility of hydraulic resonance of the runner is analyzed [18]. Yue et al. [19] calculated the static stress and mode of the reversible turbine runner under different working conditions of the turbine head, and put forward some suggestions to improve the runner structure. Wu et al. [20] analyzed the mechanism of blade crack by calculating the development mode of the mixed-flow

runner. Ma et al. [21] used a unidirectional fluid–structure coupling theory to carry out the numerical calculation of the mixed-flow runner under off-design conditions, and analyzed the potential damaged parts of the runner, which have important reference values for the study of the stability of the runner structure. Wu [22] used ANSYS 2017 software to build a 3D model of the shaft system of a pump-turbine unit as well as its electromagnetic field. The Fourier series was applied on the description of the air gap permeability to calculate the stiffness coefficient of unbalanced magnetic pull. Based on the rotor dynamics finite element method, the influence of the unbalanced magnetic pull coefficient of the generator and the stiffness coefficient of sliding bearing on the modal characteristics of the unit rotor system were deeply studied [23]. Holopainen et al. [24,25] found that the analytical expression of the unbalanced magnetic tension of shafting is the most critical parameter in structural analysis, and it is difficult to obtain. Fourier series is used to re-analyze the air gap permeability. Subbiah [26] is used to study the transient dynamic response of flexible rotors with nonlinear supports. Wang [27] used the transfer matrix method and Houbolt method to model a given rotor system in time and space, and compared the transient orbit response data with the calculated data of the finite element model.

In order to obtain the structural characteristics and modal characteristics of the shaft system of a high head pump-turbine, this paper establishes the mathematical model and calculation method of the modal characteristics of the shaft system of a pump-storage unit. According to the different connection modes and boundary conditions of the runner and the main shaft, the acoustically structural coupling method is used to analyze the modal characteristics of the pump-turbine runner. The modal characteristics and mode distribution of unit shafting are obtained.

## 2. Numerical Model

### 2.1. Power Station Basic Parameters

A pump-turbine unit of a large pumped storage power station in China is taken as the research object in this paper. NX UG12.0 software is used to build up the three-dimensional model of the runner structure, runner water body, main shaft and rotor of the unit. The runner consists of 5 long blades and 5 short blades. The three-dimensional model of the runner structure is shown in Figure 1.

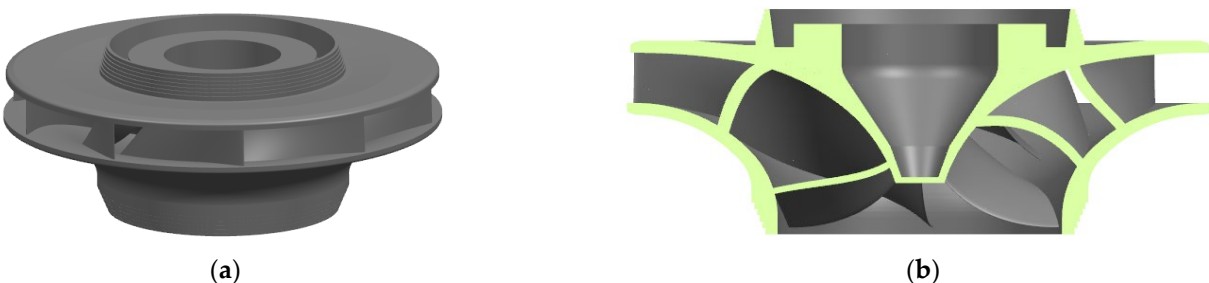

**Figure 1.** Three-dimensional model of runner structure. (**a**) Runner structure. (**b**) Cross-section diagram of runner structure.

The pump-turbine characteristic parameters represent some characteristic data in the process of converting water flow energy into mechanical energy of the turbine when water flows through the turbine, and the main shaft characteristic parameters represent the structural characteristics of the natural frequency of the turbine shaft system itself. The relevant parameters of pump turbine runner and main shaft in this study are shown in Table 1. The other boundary conditions calculated by modal analysis of the shaft system are as follows: the rotation speed is 500 r/min, and the inlet torus of the runner water model is the pressure surface.

**Table 1.** Power station characteristic parameter.

| Parameter | Value | Parameter | Value |
|---|---|---|---|
| Head H (m) | 604 | Generator rotor length $H_1$ (mm) | 3886.5 |
| Power P (MW) | 357.1 | Spindle length $H_2$ (mm) | 18,500 |
| Rotational speed n (r/min) | 500 | Runner inlet diameter $D_1$ (mm) | 4290 |
| Design flow rate Q (m³/s) | 66.6 | Runner outlet diameter $D_2$ (mm) | 2013 |
| Suction height Hs (m) | −90 | Generator rotor diameter $D_3$ (mm) | 4806 |

*2.2. Mathematical Model of Rotor Dynamics*

Modal analysis is used for the calculation of the modal parameters such as natural frequency and mode shapes of a structure, so that the structure avoids resonance, and guide engineers to predict the vibration form under different loads [20]. The free vibration equation for a single degree of freedom system is

$$0 = Kq + C\dot{q} + M\ddot{q} \tag{1}$$

where $M$ is the mass matrix of the researched structural system, $C$ is the system generalized damping matrix, $K$ is the system generalized stiffness matrix, $q$, $\dot{q}$ and $\ddot{q}$ are the displacement, acceleration and velocity vectors of the system node, and 0 is the zero matrix [28].

The vibration equation of the system under the action of continuous external excitation is

$$F = Kq + C\dot{q} + M\ddot{q} \tag{2}$$

where $F$ is the unbalanced magnetic pull of the rotor system. The unbalanced magnetic pull, as a steady-state periodic load, can be expressed as $F = (F_R \pm jF_I)e^{i\omega t}$, and $\omega_k$ is the frequency of the excitation force [29].

The governing equation of solid structure can be derived from Newton's second law:

$$\rho_s \ddot{d}_s = \nabla \cdot \sigma_s + f_s \tag{3}$$

where, $\rho_s$ is the solid structure density, $Kg/m^3$; $\ddot{d}_s$ is the local acceleration vector in the solid domain; $\sigma_s$ is the Cauchy stress tensor; $f_s$ is the volume force vector.

The acoustic-fluid–solid coupling method (acoustic-FSI) is usually used to solve the modal characteristic parameters such as natural frequency and additional mass of the structure in the water body, which uses acoustic elements to discretize the fluid domain [30]. Assuming that the fluid is compressible, non-viscous, non-rotating, and non-flowing, and that the average density and pressure remain constant throughout the flow field, the N-S and continuum equations can be simplified to the acoustic wave equation, the Helmholtz equation [31]:

$$\nabla^2 P = \frac{1}{c^2} \frac{\partial^2 P}{\partial t^2} \tag{4}$$

where, $P$ is fluid pressure, Pa; $c = \sqrt{\frac{k}{\rho_0}}$ is the speed of sound in water medium, m/s; $\rho_0$ is the average density of the fluid, $Kg/m^3$; t is time; and $\nabla^2$ is a Laplace operator.

Since there is no viscous dissipation, the Helmholtz equation is called a lossless wave equation for pressure propagation in a fluid. For the sound-fluid–solid coupling problem, Equations (3) and (4) should be considered simultaneously [32].

*2.3. Rotor System Structure and Grid Model*

Figure 2 shows the sketch of the structure of the rotor-bearing electromagnetic system (referred to as rotor system) of the researched pump turbine unit. The rotor system is mainly composed of a thrust bearing, guide bearing, generator rotor and runner. The unit adopts semi-umbrella arrangement, and the total length of the main shaft in the rotor system is about 18.5 m, the main shaft is a hollow structure, the main shaft diameter of

the upper end of the generator rotor is 0.91 m, the main shaft diameter of the lower end of the generator rotor is 1.15 m, and the main shaft diameter of the generator rotor section is 1.6 m. The maximum unbalance torque of the runner after static balance is 1.65 kg·m, and the overall material of the runner is 06Cr19Ni10 [33].

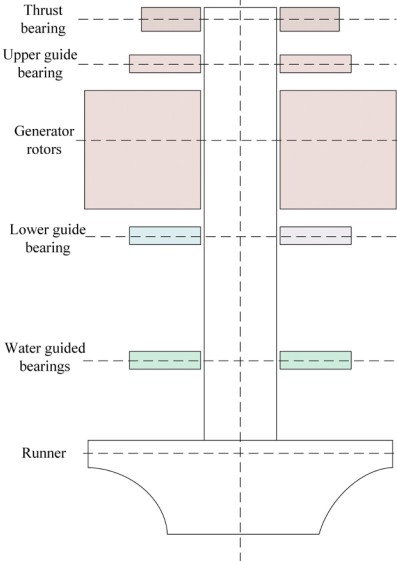

**Figure 2.** Three-dimensional structure diagram of rotor system.

The rotation system is assumed not to deform during operation of the device. Several issues that influence the behavior of the shaft are neglected such as the mass and torsional action of the shaft and thrust bearing. In this research, the bearing units are taken as the boundary condition and the rotating part is assumed as rigid. The supporting effects of three kinds of guide bearings and thrust bearings on the rotating system are simulated. For computational simplicity, only the direct rigidity of the oil film is considered, ignoring the cross-stiffness and torsional rigidity. Table 2 shows the basic parameters of the material of the researched structure.

**Table 2.** Material properties of rotor system components.

| Components | Materials | Density/(kg/m$^3$) | Modulus of Elasticity/Pa | Poisson's Ratio |
|---|---|---|---|---|
| Coils | Copper | 8900 | $1.15 \times 10^9$ | 0.33 |
| Magnetic yoke poles | Magnetic yoke | 7830 | $2.06 \times 10^9$ | 0.3 |
| Other | steel | 7850 | $2.10 \times 10^{11}$ | 0.3 |

Figure 3 shows the structure of the researched runner and the diagram of the runner–shaft connection. The connection modes of the turbine runner and the main shaft are complex. The reversible pump-turbine and the main shaft are connected through bolts and pin holes, and the upper crown end face of the turbine is in close contact with the lower end face of the main shaft. The rotational energy of the water flow is transferred to the main shaft in the form of torque through the bolts and pin holes, which drives the generator rotor to rotate and generate electric energy. At present, in order to simplify the numerical simulation and reduce the consumption of computing resources, scholars remove the bolts and pin holes between the main shaft and the runner, and adopt the bolt-free connection (contact mode is binding) between the main shaft and the runner structure as shown in Figure 3a when analyzing the modal characteristics of the shafting of the hydro-generator set.

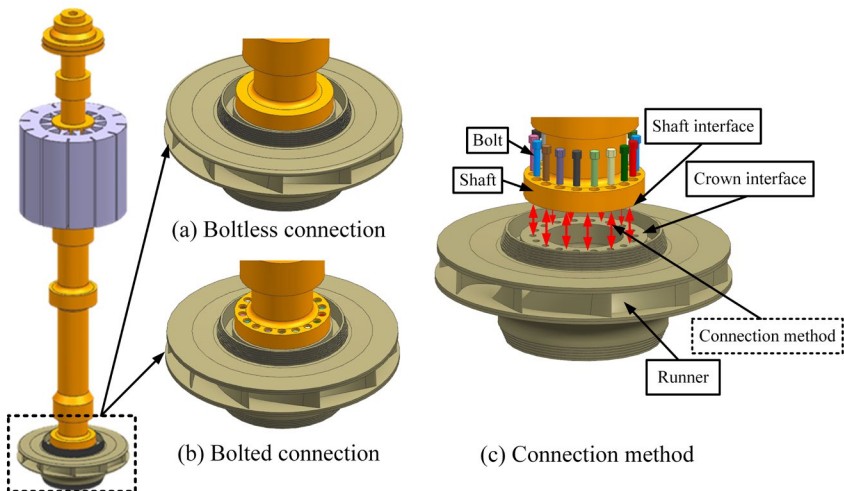

**Figure 3.** Rotor system structure model and runner-shaft connection mode diagram.

In this study, 18 bolt models are established based on the actual connection mode of the main shaft and runner, and different contact modes are adopted to calculate the modal characteristics of shafting. Table 3 shows the main connection modes of the runner and shaft.

**Table 3.** Main contact modes between runner and main shaft.

| Contact Method | Normal Separation | Tangential Slip |
| --- | --- | --- |
| Bond | Forbid | Forbid |
| No Separation | Forbid | Free |
| Rough | Free | Forbid |
| Frictionless | Free | Free |
| Frictional | Free | Free |

The 3D structure grid model of the rotor system of the pumped storage unit is shown in Figure 4. The rotor system consists of different complex parts. A structured grid is applied for the circumferential distributed components in the rotating subsystem while an unstructured grid is applied for the runner. In addition, grid encryption is processed and adjusted for the small structural plane in the rotating subsystem in order to ensure the mesh quality and the accuracy of the grid node connection.

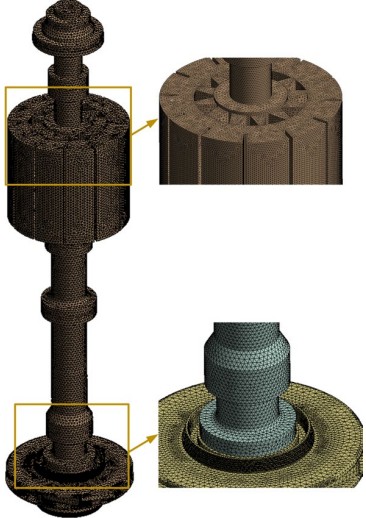

**Figure 4.** Three-dimensional structure grid model of rotor system.

Considering the accuracy of numerical calculation results and the comprehensive consideration of computer resources, this paper takes the natural frequency of the shaft system as the goal to verify the changes of the natural frequency of the first four modes of the grid model of the rotor system with different grid numbers. Finally, the calculation model of the rotor system with grid numbers of $4.09 \times 10^6$ is selected. The grid independence check has been performed and the result is shown in Figure 5.

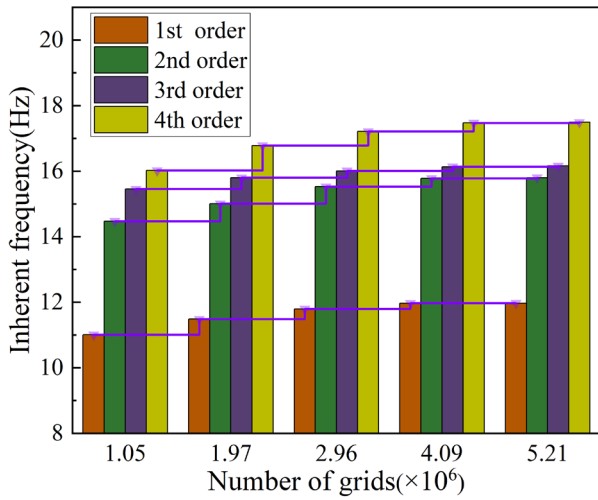

**Figure 5.** Grid-independent verification of rotor systems.

## 3. Result and Discussion

### 3.1. Numerical Simulation of Working Conditions

In order to explore the modal characteristics of a turbine runner in a pumped storage power station under different connection modes, this paper mainly takes the main shaft and runner as the main research objects, studies two models of a single runner and a runner with main shaft, respectively, considers different types of contact modes between the main shaft and runner, and develops eight different working conditions. When the lower end face of the main shaft and the upper crown end face of the runner are bolted, the contact type is friction, and the friction coefficients are 0.1 and 0.6, respectively. Specific working conditions are shown in Table 4.

**Table 4.** Numerical simulation of working conditions.

| Number | Conditions | Connect Type | |
|---|---|---|---|
| 1 | Single runner | Free | |
| 2 | Single runner (Fixed) | Free | |
| 3 | Runner + axis (boltless connection) | Bond | |
| 4 | | Bond | |
| 5 | | No separation | |
| 6 | Runner + axis (bolted connection) | Rough | |
| 7 | | Fricional | 0.1 |
| 8 | | | 0.6 |

### 3.2. Modal Analysis of Runner

The turbine shaft system of the hydropower station mainly includes runner, main shaft and generator rotor, and the main shaft and turbine runner are usually connected by bolts. The bolt transfers the rotating torque of the runner to the main shaft and ultimately generates electrical energy. In this study, the bolts are bound on the main shaft and the runner, respectively, and the interface of the main shaft and the runner adopts a rough contact type. Table 5 shows a comparison of the natural frequencies of the runner in air and water.

**Table 5.** Comparison of natural frequencies of the runner in air and water.

| Modal Shape | Air Mode with Shaft (Hz) | Water Mode with Shaft (Hz) | Frequency Change Rate |
|---|---|---|---|
| 1ND | 121.48 | 48.807 | −59.82% |
| (0,1) | 172.86 | 61.749 | −64.28% |
| 2ND | 195.59 | 120.07 | −38.61% |
| 3ND | 299.65 | 195.87 | −34.63% |
| 4ND | 363.91 | 253.66 | −30.30% |
| 5ND | 388.29 | 313.35 | −19.30% |
| 6ND | 519.79 | 328.61 | −36.78% |

Due to the unique geometric characteristics of the pump-turbine runner, the axial stiffness of the upper crown and lower band structure of the runner is lower than the axial stiffness of the runner blade, and the upper crown and lower band structure at the front end of the runner inlet lacks blade support, so the axial stiffness is significantly reduced, and the structure is easily deformed. The upper crown and lower band structure of the runner are similar to that of the disk, so the mode of the runner is also similar to that of the disk. Each mode can be distinguished by the number of pitch circles (NC) and pitch diameters (ND). In this paper, vibration sectors are divided in the circular direction of the runner according to the node-diameter number ND to describe the vibration pattern. Figure 6 shows the vibration pattern of the runner with water mode 1ND to 6ND according to the natural frequency. The vibration pattern of 1ND represents axial vibration around a line with a displacement of 0, and the vibration pattern of (0,1) represents the axial vibration of the runner as a whole. The 2ND, 3ND and 4ND modes are similar, where the stiffness of the low-pressure side of the upper crown of the runner is small, so the vibration amplitude of this region in each sector is large, especially the low-pressure side of the upper crown. By comparing the air and water modes of the runner, it is found that the two modes are similar, but the vibration amplitude in the water mode is smaller than that in the air mode, which indicates that the water medium has little influence on the vibration mode of the runner, but will significantly reduce the amplitude of the runner. Reference [22] reveals the influence of the unbalanced magnetic pull on the modal characteristics of the shaft system, and the results show that the difference of natural frequencies of the reversible pump-turbine runner in dry mode and wet mode is consistent with the research results in this paper.

Figure 7 shows a comparison on the natural frequencies of the runner in water and air. It can be seen from the diagram that, compared with the natural frequency of the runner in air, the natural frequency of the runner in water is greatly reduced under the influence of additional mass, and the frequency reduction rate of the (0,1) mode can reach more than 64%.

Air                       Water

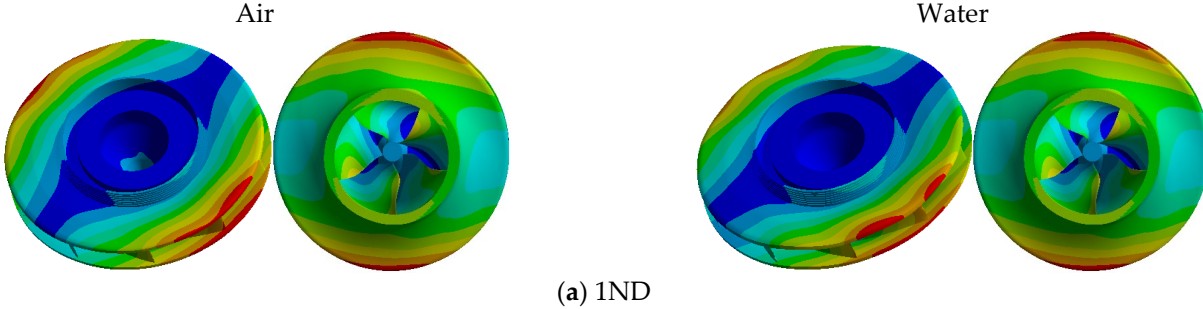

(**a**) 1ND

**Figure 6.** *Cont.*

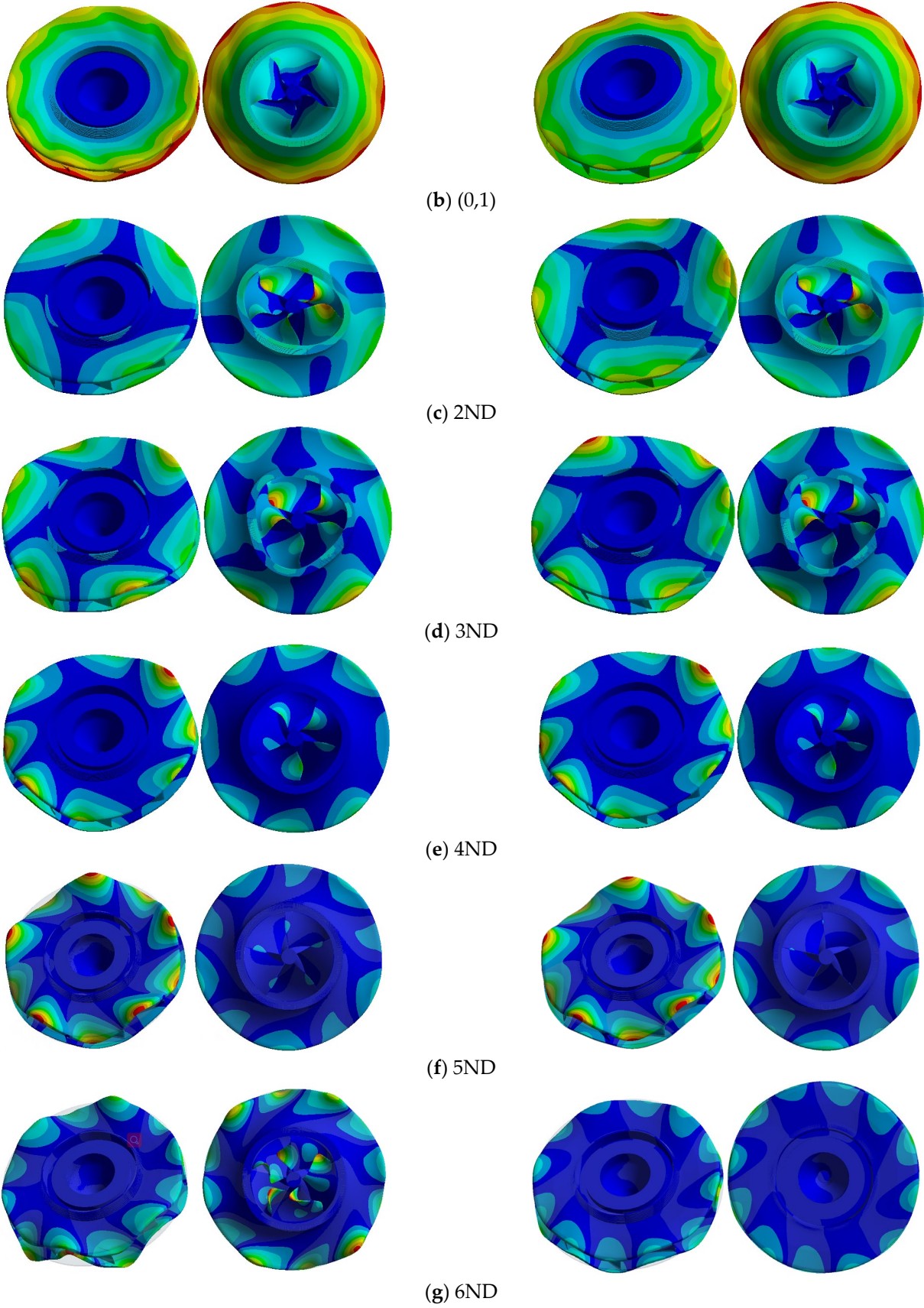

(**b**) (0,1)

(**c**) 2ND

(**d**) 3ND

(**e**) 4ND

(**f**) 5ND

(**g**) 6ND

**Figure 6.** Runner modes in air and water.

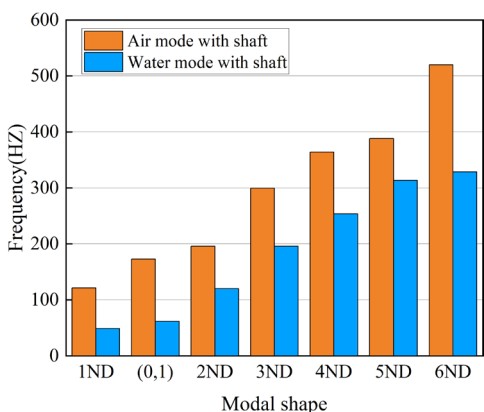

**Figure 7.** Comparison of 1ND–6ND natural frequencies of runner in air and water.

In air mode and water mode, the results of runner mode calculation with and without shaft boundary conditions are compared, the influence of runner mode calculation method on the calculation results is analyzed and evaluated, and the accuracy of the simplified calculation method is verified. For the modal calculation of the runner with a shaft, spring constraints are applied to the upper guide bearings, lower guide bearings, water guide bearings and thrust bearings. For the modal calculation of the shaftless runner, a fixed constraint is applied to the runner–shaft connection surface. Table 6 shows the natural frequencies of the runner in water mode and air mode under two calculation methods, with and without a shaft. Reference [34] confirms that when the pump turbine runner is connected to the main shaft, the stiffness of the whole shaft system will change, and the natural frequency in the low-order mode will have a greater impact, while the natural frequency in the high-order mode will have a lesser impact.

**Table 6.** The natural frequency of the runner calculated by two modal calculation methods, with and without shaft.

| Modal Shape | Air Mode with Shaft (Hz) | Air Mode without Shaft (Hz) | Frequency Change Rate | Water Mode with Shaft (Hz) | Water Mode without Shaft (Hz) | Frequency Change Rate |
|---|---|---|---|---|---|---|
| 1ND | 121.48 | 120.17 | −1.08% | 48.807 | 69.138 | 41.66% |
| (0,1) | 172.86 | 169.73 | −1.81% | 61.749 | 77.872 | 26.11% |
| 2ND | 195.59 | 193.78 | −0.93% | 120.07 | 123.41 | 2.78% |
| 3ND | 299.65 | 296.02 | −1.21% | 195.87 | 197.25 | 0.70% |
| 4ND | 363.91 | 358.46 | −1.50% | 253.66 | 254.76 | 0.43% |
| 5ND | 388.29 | 382.07 | −1.60% | 313.35 | 328.27 | 4.76% |
| 6ND | 519.79 | 513.42 | −1.23% | 328.61 | 341.57 | 3.94% |

Figure 8 shows the comparison of the natural frequencies of the air and water modes of the runner calculated by two modal calculation methods with and without an axis in the form of a graph. For the air mode calculation, the runner mode frequency obtained by the axial method is close to that obtained by the axial method, while the runner mode frequency obtained by the axial method is slightly lower, and the frequency change rate is less than 2%. For the calculation of the water mode, the frequency of the runner mode calculated by the shaft-free method is slightly increased, and the frequency change rate is slightly larger than that of the air mode. The maximum change rate occurs in the 1ND mode, reaching 41.66%. For the 2ND and higher natural frequencies, the frequency change rate of the shaft-free method is less than 4%. On the whole, the error of the simplified calculation method is less than 4% for the modal frequencies 2ND and above. In reference [34], it is suggested that the blade vibration modes of runner blades with and without shafts are consistent, while the stiffness is inconsistent. This is consistent with the results of this paper.

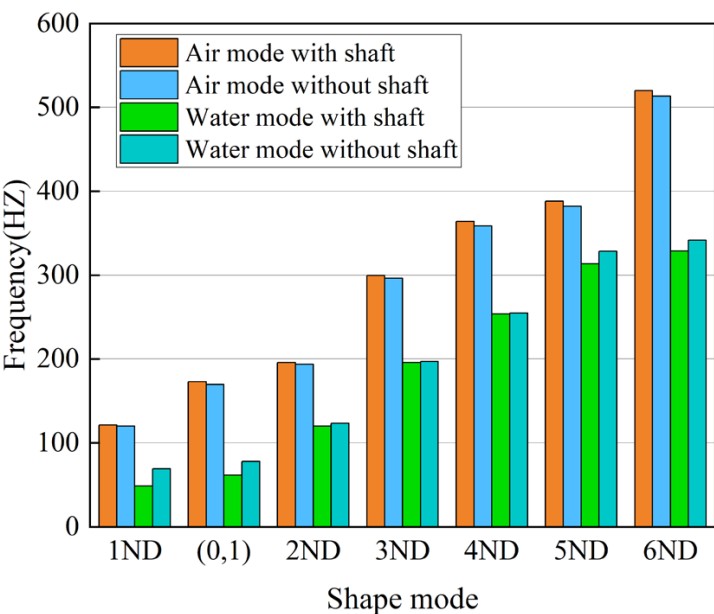

**Figure 8.** Comparison of natural frequency of air and water modes of runner calculated by two modes calculation methods with and without shaft.

### 3.3. Influence of Different Contact Modes on Modal Characteristics of Runner

The main shaft and the upper crown are connected by bolts. The upper part of the bolt is bound to the main shaft, and the lower part of the bolt is bound to the upper crown of the runner. In modal acoustic calculation, the lower end face of the main shaft and the upper end face of the rotor will produce relative displacement when the main shaft rotates. The water mode of shafting is numerically calculated for different types of contact modes, and the modal characteristics and natural frequencies of shafting are obtained by setting different contact types.

Table 7 shows the modal natural frequencies of shafting under different contact types of the main shaft and runner of the pump-turbine. The results show that the natural frequency of the running wheel in the water body will be reduced by 19% to 64% under the damping effect of the water body medium. However, the shafting of the pump-turbine unit has a huge mass, and different contact types have little influence on the natural frequency of the shafting. The natural frequency reduction rate of each order mode of the runner is less than 0.1%, and the frequency reduction number of each order mode is related to the vibration mode of the runner. It can be seen that the interaction between the runner and the water improves the vibration energy and makes the runner more prone to vibration when the runner works in the water medium. Reference [35] confirms that several different constraints of the pump turbine shaft system will produce different results for modal analysis. It is proved in this paper that the surface contact condition of the runner and spindle has little effect on the mode of the shaft system.

**Table 7.** Natural frequencies of runner modes under different contact types.

| Modal Shape | Bond (Hz) | No Separation (Hz) | Rough (Hz) | Fricional (Hz) | |
| --- | --- | --- | --- | --- | --- |
| | | | | *f* = 0.1 | *f* = 0.6 |
| 1ND | 48.807 | 48.779 | 48.745 | 48.762 | 48.76 |
| (0,1) | 61.749 | 61.688 | 61.701 | 61.699 | 61.697 |
| 2ND | 120.07 | 119.66 | 119.77 | 119.72 | 119.71 |
| 3ND | 195.87 | 195.81 | 195.62 | 195.70 | 195.71 |
| 4ND | 253.66 | 253.25 | 253.12 | 253.19 | 253.18 |
| 5ND | 313.35 | 312.99 | 312.52 | 312.76 | 312.76 |
| 6ND | 328.61 | 328.42 | 328.29 | 328.31 | 328.33 |

Figure 9 shows the modal characteristic distribution of the shafting model under a rough contact type. The 1ND mode is manifested as the transverse oscillation of the shafting

structure. The (0,1) mode and the higher ND mode mainly occur in the runner structure. For the ND mode, the frequency decline rate decreases with the increase in the order. (0,1) The motion direction of each part of the upper crown (lower band) is the same. For the ND mode, when one part of the upper crown (lower band) moves upward, the other part will move downward. In this way, the water body displaced when one part of the runner moves upward can supplement the missing water body after the other part moves downward, so the additional mass coefficient of the ND mode is lower. Reference [36] confirms that the vibration modes in the low order mode of the rotating wheel shaft system mainly occur on the runner structure, and the shaft system structure is affected by the change of flow field, and verifies the authenticity and rationality of the wet mode calculation method in this study in line with the calculation of the shaft system structure.

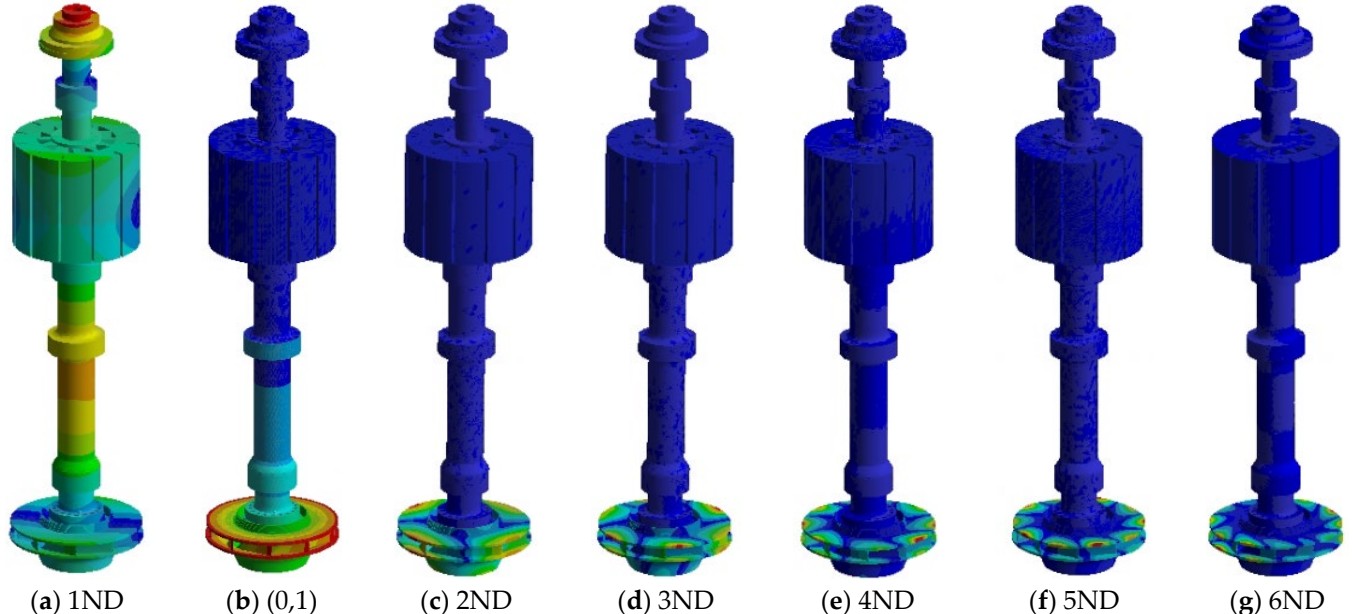

| (**a**) 1ND | (**b**) (0,1) | (**c**) 2ND | (**d**) 3ND | (**e**) 4ND | (**f**) 5ND | (**g**) 6ND |

**Figure 9.** Modal shapes of shafting under rough contact surface.

## 4. Conclusions

In this paper, the dynamic characteristics of a pump-turbine runner has been simulated by means of different types of simplified models. The finite element method results from different models have been obtained and compared and the influence of different border condition factors have been evaluated. The main conclusions are listed below:

1. Affected by the additional mass of the water body, the natural frequency of the runner in the water is greatly decreased, and its decrease rate is between 19% and 64%. Comparing the results of simplified calculation with and without a shaft, it is found that the simplified calculation method has a great influence on the lower order mode, and the calculation error is less than 4% for the 2ND and above modes.
2. When the spindle and runner are connected by bolts, the contact surface of the two has little influence on the natural frequency of shafting. The natural frequency difference of each order mode under the three contact modes of "No separation", "Rough" and "Frictional" is less than 0.2%.
3. The results of numerical simulation can help the power station to improve the structure of the shaft system, ensure the safe and stable operation of the power station, improve the operation life and economic benefits of the power station, and ensure the high efficiency of the hydropower project and the sustainable and circular development of energy.



**Author Contributions:** X.L., J.P., L.L., W.Z. and Z.W. proposed the simulation method. J.P., W.Z., X.L. and L.L. completed the numerical simulations. X.L., J.P., W.Z. and D.Y. analyzed the data and wrote the paper. J.P. and Z.W. revised and reviewed the paper. Y.W. and L.Z. provided some data support for the study. All authors have read and agreed to the published version of the manuscript.

**Funding:** The authors would like to express their sincere thanks for the financial support of the project: Evaluation and Calculation Service for the Coupling Performance of the Shaft System of Pump-turbine Unit in Luoning Pumped Storage Power Station of State Grid Xinyuan Group Co., Ltd. (approval No. 20222001266). The research is funded by China Postdoctoral Science Foundation (Grant No. 2022M711768).

**Informed Consent Statement:** Not applicable.

**Data Availability Statement:** The data presented in this study are available on request from the corresponding author. The data are not publicly available because they also form part of an ongoing study.

**Conflicts of Interest:** The authors declare no conflict of interest.

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
