# Peer review of "Comparative Study on Numerical Calculation of Modal Characteristics of Pump-Turbine Shaft System"

_jmse, doi:10.3390/jmse11112068_

Round 1
Reviewer 1 Report
In your study, you mentioned the influence of different contact modes between the main shaft and the runner on the modal characteristics. Could you provide some examples of these contact modes and their corresponding effects on system behavior?
How does the frequency and amplitude of the resonance phenomena in pumped storage units impact the long-term durability and reliability of the runner and shaft components? Are there specific maintenance or inspection protocols that should be considered in light of your findings?
Can you discuss any practical challenges or considerations associated with implementing the optimization of the analysis algorithm for runner structure in real-world applications of large pumped storage units? Are there any computational or data acquisition requirements that need to be addressed?
Beyond the impact on structural integrity, did your research explore the potential effects of resonance on the overall efficiency and performance of pumped storage power stations? How might resonance-induced stress and strain affect the energy conversion process or the lifespan of these systems?
Given the global shift toward renewable energy sources, how do your research findings contribute to the broader goal of improving the reliability and efficiency of pumped storage power stations as a means of storing and delivering renewable energy? Are there policy or industry implications for promoting the adoption of such technologies based on your work?
Are there specific types of sensors or monitoring systems that you would recommend for real-time monitoring of runner and shaft behavior in pumped storage units, particularly in the context of resonance detection and prevention?
Could you discuss any potential interdisciplinary collaborations that your research might enable? For instance, how might your findings be relevant to the fields of materials science, structural engineering, or renewable energy technology beyond the scope of fluid dynamics and mechanics?
In practical terms, how frequently should pumped storage power stations undergo structural assessments or maintenance procedures based on the insights gained from your research? Are there guidelines or best practices that operators should follow to ensure the long-term reliability of these systems?
Did your research explore any innovative materials or engineering techniques that could be applied to mitigate resonance and stress-related issues in runner and shaft components? Are there emerging technologies that hold promise in this regard?
Finally, how do you envision the future of research in this field? Are there specific avenues for further investigation or emerging challenges that researchers should prioritize based on the findings and implications of your work?
Reviewer 2 Report
This work aimed to study the influence of different connection modes and boundary conditions on the shafting modal characteristics of a reversible pump-turbine runner, the modal characteristics and mode distribution of the shafting are obtained by means of acousto structural coupling method. However, it is within the scope of the JMSE but it needs sensible effort and major revisions, as follows:
1. This topic has been investigated many times in literature; many theoretical and experimental published research papers on Hydraulic turbine have not been cited to see the difference between your work and others, and this is contrary to the ethics of scientific research.
2. Title: Please, use the literature background to develop a better title in order to increase catch the prospective reader's attention.
3. Abstract: It should be rewritten in terms of aim, background, motivation, and significant results. Your abstract should clearly state the essence of the problem you are addressing, what you did and what you found and recommend. That would help prospective readers of the abstract to decide if they wish to read the entire article.
4. Introduction: it is not well organized. The mentioned description of each piece of literature should be shortened and instead of that give a short detail about what did in this literature. The last paragraph should be rewritten by highlighting the main contribution of current work compared with existing literature. The main concern I have about the paper is with respect to the contributions of your work. The used methodology has a few outstanding innovation points.
5. In section 2: system description should be presented for each component. Assumptions, operating conditions range and values, and all input data should be presented in this section.
6. Also, preliminary design sub-section should be added in terms of dimensions and configuration equations with solution procedure, operating conditions with input data for turbine design should be presented in clear way. More well-known information has been provided and the newly developed methodology is not clear, what is the difference between the current approach and the existing one? What is the contribution/effort in the developed approach that is missing?
7. Boundary conditions and input data should be tabulated and presented in clear way.
8. Moreover, the CFD setup through the ANSYS workbench should be provided in a clear way. More justification about the selection of turbulence model should be provided.
9. In section 3, validation of the results and comparison to the results by published open literature is not clear. This section should be rewritten to present the validation in a clear way; without considering this comment, the manuscript cannot be accepted.
10. In results and discussion section: the impact of the developed pump turbine design methodology should be presented in terms of design parameters on the turbine performance i.e. turbine efficiency and power output and then use this performance to recalculate the system thermal efficiency. Such as total expansion ratio, number of blades, hub to tip ratio, tip clearance, … etc. please, the mentioned literature, and others.
11. Discussion of the results should provide useful in-sights. Where, the results should be further elaborated to display how they could be utilized in real uses. The authors should further grow critical assessment in results discussion.
12. The work does not provide a well-written conclusion section in terms of main findings and contribution. Relevance of the work with respect to the energy sustainability aspect should be discussed in the conclusion section.
English should be improved.
Round 2
Reviewer 2 Report
The revised version of the manuscript (jmse-2569415) showed the authors made an effort to improve. The original manuscript was revised according to raised comments and suggestions by the reviewer. The reviewer’s questions were answered properly and the overall quality of the paper has improved after the revision. Based on the changes made, it can be accepted.